# Neutrophil Extracellular Traps-DNase Balance and Autoimmunity

**DOI:** 10.3390/cells10102667

**Published:** 2021-10-05

**Authors:** Andrea Angeletti, Stefano Volpi, Maurizio Bruschi, Francesca Lugani, Augusto Vaglio, Marco Prunotto, Marco Gattorno, Francesca Schena, Enrico Verrina, Angelo Ravelli, Gian Marco Ghiggeri

**Affiliations:** 1Division of Nephrology, Dialysis and Transplantation, IRCCS Istituto Giannina Gaslini, GenoaLargo Gaslini, 16148 Genoa, Italy; andreaangeletti@gaslini.org (A.A.); enricoverrina@gaslini.org (E.V.); 2Laboratory of Molecular Nephrology, IRCCS Istituto Giannina Gaslini, GenoaLargo Gaslini, 16148 Genoa, Italy; mauriziobruschi@yahoo.it (M.B.); francescalugani@gaslini.org (F.L.); 3Center for Autoinflammatory Diseases and Immunodeficiencies, IRCCS Istituto Giannina Gaslini, 16147 Genoa, Italy; stefanovolpi@gaslini.org (S.V.); marcogattorno@gaslini.org (M.G.); francescaschena@gaslini.org (F.S.); angeloravelli@gaslini.org (A.R.); 4Dipartimento di Neuroscienze, Riabilitazione, Oftalmologia, Genetica e Scienze Materno Infantili, University of Genoa, 16132 Genoa, Italy; 5Department of Biomedical, Experimental and Clinical Sciences “Mario Serio”, University of Firenze, 50121 Firenze, Italy; augusto.vaglio@virgilio.it; 6Institute of Pharmaceutical Sciences of Western Switzerland, School of Pharmaceutical Sciences, University of Geneva, 1205 Geneva, Switzerland; Marco.Prunotto@unige.ch; 7Clinics of Pediatrics and Rheumatology, IRCCS Istituto Giannina Gaslini, 16147 Genova, Italy

**Keywords:** neutrophil extracellular traps, systemic lupus erythematosus, lupus nephritis, autoimmune disease, immunosuppressive treatment

## Abstract

Neutrophil extracellular traps (NETs) are macromolecular structures programmed to trap circulating bacteria and viruses. The accumulation of NETs in the circulation correlates with the formation of anti-double-stranded (ds) DNA antibodies and is considered a causative factor for systemic lupus erythematosus (SLE). The digestion of DNA by DNase1 and DNases1L3 is the rate- limiting factor for NET accumulation. Mutations occurring in one of these two DNase genes determine anti-DNA formation and are associated with severe Lupus-like syndromes and lupus nephritis (LN). A second mechanism that may lead to DNase functional impairment is the presence of circulating DNase inhibitors in patients with low DNase activity, or the generation of anti-DNase antibodies. This phenomenon has been described in a relevant number of patients with SLE and may represent an important mechanism determining autoimmunity flares. On the basis of the reviewed studies, it is tempting to suppose that the blockade or selective depletion of anti-DNase autoantibodies could represent a potential novel therapeutic approach to prevent or halt SLE and LN. In general, strategies aimed at reducing NET formation might have a similar impact on the progression of SLE and LN.

## 1. Introduction

The formation of extracellular traps by neutrophils (or NETs) is part of the innate immune response and consists of the release of DNA from neutrophils and granule components that, once outside the cell, compose a net where pathogens are entrapped and killed through proteolytic mechanisms [1,2,3]. The activation of nicotinamide-adenine-dinucleotide-phosphate (NADPH) oxidase is linked to the generation of NETs and the activation of intracellular granular proteases [3]. The complex and interactive network of molecules activated during NETosis is part of the initial immune response against any type of infection [4]. Indeed, subjects affected by inherited disorders causing the inactivation of NADPH oxidase, such as chronic granulomatous disease, are more exposed to bacterial and fungal infections [4,5].

Considering the elaborate structure involving chromatin-DNA and more than 300 proteins [6], and given the interactive nature of the functions, the significance of NETs goes beyond the immune response [3]. In the last decade, consolidated evidence has demonstrated that DNA, and proteins derived from NETs, may serve as autoantigens in several autoimmune diseases [6,7]. The complex of DNA and oxidized proteins acts, in fact, as a hapten, stimulating the formation of autoantibodies more intensely than DNA or proteins alone [6]. The link of NETs with autoimmunity is particularly evident in the context of systemic lupus erythematosus (SLE) and lupus nephritis (LN), since NETs represent an important source of the two major antigens in both conditions [8]: DNA and oxidized (93 methionine sulfoxide) α-enolase. Studies measuring NET levels in SLE and LN suggest the relevance of maintaining a physiological balance between formation and removal that is critical for reducing the formation of autoantibodies in both conditions [9,10].

## 2. NET Levels and Formation in Autoimmune Conditions

Neutrophil-generating NETs, also known as NET remnants, can be detected in circulation through an ELISA test specific for myeloperoxidase (MPO) and, therefore, able to detect the DNA–MPO complex of NETs [8].

In the last two decades, on the basis of this assay, several studies have reported increased circulating NETs in subjects affected by autoimmune conditions, such as small vessel vasculitis [11,12], and SLE/LN [10,13,14]. This finding does not necessarily mean that NET production is increased in autoimmunity. In fact, direct evidence for an increased production of NETs in any of the clinical settings above-mentioned is lacking. The unique indirect evidence is that neutrophils derived from patients with SLE/LN, and stimulated with phorbol 12-myristate 13-acetate (PMA), produce more and different NETs compared to neutrophils derived from healthy subjects [15]. When PMA was infused in rats to stimulate NETs, the rodents developed a sort of pulmonary capillaritis, miming the small vessel vasculitis associated with anti-MPO autoantibodies [15].

In a similar way, neutrophils from the circulation of New Zealand mice, a model of spontaneous lupus, are able to produce an increased formation of NETs compared to neutrophils derived from control mice [16].

## 3. NET Balance in Systemic Lupus Erythematosus

The increased NET production in autoimmunity, as reported above, is of interest and represents a possible mechanism. On the other hand, several findings indicate that, in SLE, increased NETs may result from reduced degradation rather than increased production [3]. Taken together, these studies suggest that the balance between NET production and removal plays a critical role in SLE and other autoimmune conditions. NET removal is, accordingly, crucial to maintaining the right balance between NET formation and degradation. Of most importance, it was shown that the entity of reduction covaried with disease activity. In particular, patients with a reduced ability to remove NETs had lower levels of the circulating complement components, C3 and C4 [17,18] that, when reduced, represent the common clinical markers of increased disease activity. Moreover, such subjects presented increased circulating levels of anti-DNA and anti-histone antibodies and developed, in many cases, glomerulonephritis [9]. The laboratory approach, in the first series of studies, was based on testing the capability of the sera, obtained prospectively from patients with SLE, to remove in-vitro-generated NETs and, therefore, did not focus on the possible mechanisms. As a main result of the initial functional studies, DNases emerged as fundamental in removing NETs [9], and a strong association between the reduction of DNases activity and the accumulation of NETs in autoimmune conditions was reported [9].

The DNase complex is composed of three enzymes, DNase I, DNase II, and DNase1L3, with roles in the digestion and removal of circulating DNA. They have specificities for different DNA and are variably implicated in maintaining a correct DNA balance. In LN, the removal of DNA, and consequently of NETs, may be impaired for different reasons [19]. One reason is the loss-of-function mutations in one of the genes coding for the DNases [20,21,22]. A second mechanism that may lead to DNase functional impairment is the presence of DNase inhibitors in the sera of patients with low DNase activity [9], or the generation of anti-DNase antibodies [9,23]. This phenomenon has been described in a significant number of patients, and may actually represent a relevant mechanism determining increased levels of NETs in a significant number of subjects affected by LN [24].

## 4. Circulating DNA Forms and DNase Specificity

As mentioned, the presence of extracellular DNA, frequently in association with multiple proteins [8], is critical for the anti-DNA antibody generation process and is intimately associated with the different extracellular DNA species. To further increase complexity, DNase acting upon those DNA species might well modulate the anti-DNA antibody-generation process. Below, we review the literature related to both topics.

Extracellular DNA may be defined based on physical characteristics, such as variable size, varying from short naked DNA to DNA as part of a chromatin strand, and follows, in each case, specific degrading pathways. The nucleosome is, hierarchically, the largest structure containing DNA. It corresponds to the basic unit of chromatin and is formed by a framework of Histone 2A, 2B, 3, and 4 assembled as an octamer, surrounded and wrapped by DNA.

Nucleosomes are generated during cell apoptosis by chromatin cleavage. In SLE, specific antinucleosomes are directed towards conformational epitopes created by the interaction between dsDNA and the core histones. Moreover, nonspecific antinucleosome antibodies recognize the basic elements of the nucleosome: the histones and the DNA [25]. In the last two decades, nucleosomes have emerged as the principal antigen in the pathophysiology of SLE, and antinucleosome antibodies are closely associated with organ damage [26,27]. Nucleosomes have been shown to be more strongly immunogenic than native DNA or histones, and induce a strong T-helper-cell response [28]. Moreover, antinucleosome antibodies were recently proposed as a marker to identify patients with a higher risk of developing renal relapse in inactive SLE [29,30]. It is largely recognized that the physical form and the length of DNA are directly correlated and may determine its antigenicity. The formation of antibodies against naked DNA develop later than antibodies versus protein-bound DNA, suggesting that the whole complex of hapten-DNA, rather than its individual components, is mainly involved in breaking the immunotolerance [31]. Moreover, longer fragments of DNA, due to a more extended bivalent surface, have increased avidity for anti-dsDNA antibodies [31,32].

Chromatin may exist as small soluble fragments, or as larger extracellular structures derived from cells, such as NETs [33], or microparticles (MP) derived from apoptotic cells [34,35,36]. The removal of extracellular DNA by DNase I and DNase1L3 represents the critical step in DNA metabolism [37]. DNase I preferentially digests naked cell-free DNA, while chromatin and MP-bound-chromatin DNA are degraded by DNase 1L3 [19,38]. While in healthy conditions, a variable amount of extracellular DNA (20–50%) is transported by MP, recent findings report that the fraction enriched in longer fragments may be significantly increased in LN patients with reduced DNASE 1L3 activity [39]. A third form of intracellular DNase, DNase II, is responsible for the degradation of DNA from apoptotic bodies. Overall, DNase activity is reduced in the serum of SLE/LN patients, while circulating DNase I levels are normal, suggesting that DNase 1L3-serum-level modification is directly responsible for the reduced DNase activity [10], determining the imbalance in extracellular DNA responsible for anti-ds DNA production.

Furthermore, dendritic cells and macrophages produce the large amount of circulating DNASE1L3, supporting the fundamental role of these cells in maintaining self-tolerance and protection from autoimmunity [40,41].

## 5. DNase Mutations and Monogenic SLE

Deletions or mutations of any of the *DNASE* genes are inevitably associated with immunologic syndromes, with the common involvement of the kidney, phenotypically characterized by an autoimmune glomerulonephritis. In vivo studies using *DNASE*-knocked-out mice confirmed the direct correlation between DNase activity and autoimmune disease [31].

Mutations in exon 2 of *DNASE1* have been described in 2001, by Yasutomo, in two patients with SLE [16]. As expected from the presence of a stop codon in the *DNASE1* sequence, both patients had low levels of circulating DNase I and high levels of anti-DNA antibodies. Supporting that hypothesis, the genetic deletion of DNase I in vivo results in serological features resembling those in SLE patients, with subsequent renal involvement in the form of an autoimmune glomerulonephritis characterized by IgG and C3 glomerular deposition [42].

Bi-allelic mutations in *DNASE2* have been reported in three children who presented the same clinical phenotype, characterized by recurrent febrile episodes, fibrosing hepatitis, and membranoproliferative glomerulonephritis [17]. The serum levels of anti-DNA antibodies were fluctuant, and none of the children fulfilled the clinical criteria of SLE. However, as a common feature, a significantly high type I interferon signature was reported, suggesting the inclusion of this syndrome in the interferon-mediated inflammatory diseases that also characterize SLE.

Homozygous null mutations of *DNASEIL3* cause the pediatric onset of familial SLE that is characterized by high levels of circulating anti-dsDNA antibodies and renal involvement [18]. Clinical variability may also exist and, in a few families, the disease initially manifests as hypocomplementemic urticarial vasculitis syndrome (HUVS) [43,44] that may progress, in surviving members, to severe SLE. In the same way, a polymorphism of *DNASE1L3 (rs35677470)* coding for an R206C [45] amino acid substitution is associated with less severe autoimmune diseases, including SLE, scleroderma, and rheumatoid arthritis.

The available literature demonstrates the inverse correlation between circulating DNase1L3 and the formation of antichromatin and anti-dsDNA antibodies, with consequent clinically relevant SLE-like disease and renal involvement [19,36,42]. *DNASE1L3*-deficient mice develop a typical lupus syndrome [19], and have been widely used to support a direct implication of DNASEIL3 in SLE/LN.

Overall, mutations of any *DNASEs,* even rare, are always associated with an inflammatory syndrome with profound clinical impact that evolves, in the majority of cases, to SLE and LN.

## 6. DNase Inhibitors and Anti-DNase Antibodies in Lupus Nephritis

A decade ago, Hakkim et al. [11] first focused on the central role of DNase I for disassembling NETs, and then correlated the functional impairments of DNase I with the impaired degradation of NETs in a subset of patients with SLE. They further showed that, in some subjects, defined as ‘non degraders’, a physiological NET balance was restored by removing serum antibodies or by adding the sera of a healthy donor [11]. On the basis of these findings, they postulated the existence of anti-DNase I antibodies or, alternatively, of DNases I inhibitors in the sera of SLE patients that correlated with disease activity and with progression to LN [9].

The second confirmatory study of the presence of anti-DNase antibodies that interfere with NET degradation was described in subjects affected by MPO-ANCA-associated microscopic polyangiitis (MPA) [46]. The authors describe a lower DNase I activity in patients than in the healthy controls, and demonstrate that IgG depletion from MPO-ANCA-associated MPA sera partially restores NET degradation. Finally, the addition of DNase I synergistically enhanced this restoration [35].

More recently, Bruschi et al. [10] found that circulating NET levels were high in 216 incident SLE patients, half of which had incident LN, and correlated with either high anti-dsDNA antibody-circulating levels or low C3 activity. DNase activity was found to be selectively decreased in patients with LN compared to patients with SLE and the controls, despite similar serum levels of DNASE 1. A total of 20% of LN patients had a 50% reduction in DNase activity. In these cases, the pretreatment of the serum with Protein A restored DNase efficiency, implying the presence of an inhibitory immunoglobulin in the plasma of patients with LN.

More recently, Hartl et al. [39] provided evidence for the direct implication of anti-DNase antibodies in SLE complicated by different organ pathologies. They performed a reliable assay for circulating DNase1L3 activity and found low levels in 50% of patients with LN compared to patients with uncomplicated SLE and the healthy controls. In LN, DNase1L3 activity was lower in those patients with active proteinuria compared to those in remission. Since DNASE 1L3 genetic deficiencies are quite rare, and could not account for the reduced DNase1L3 activity in half of the patients, an autoimmune mechanism was postulated [39]. The same authors tested whether the autoantibodies to DNase 1L3 might contribute to decreased activity [39] and found the high and specific binding of IgG to DNase 1L3 in the plasma of patients with LN correlating with activity; on the other hand, no binding to DNase I was observed. Overall, the findings by Hartl et al. [39] support the mechanistic hypothesis that the formation of anti-DNase 1L3 antibodies mediates the inhibition of its activity in patients with LN. As a consequence, the increase of polynucleosome MP-bound DNA corresponds with the high-antigenic DNA that mediates antibody formation.

## 7. Potential Treatments

The modulation of either the NET production or the DNA removal appear as two possible effective strategies in SLE/LN treatment, and a balance of the two approaches may better produce positive effects.

Blocking NET production is still an experimental area of investigation that has been recently reviewed in detail [3]. However, blocking NET production may fail and, in some cases, it impacted negatively on the general clinical status for the onset of severe complications [3]. The development of new drugs are still at the preliminary conceptual phase and will require more time for effective development [47]. Modulating the DNase activity seems to represent a more concrete opportunity, especially in secondary SLE, and it may be achieved by removing or blocking the synthesis of the circulating inhibitory substances of such enzymes.

On the other hand, plasmapheresis presents a valuable opportunity, with the aim of blocking the overall autoantibody production with the consequent relevant immune depression. Plasmapheresis has been widely used in the past; however, efficacy has only been supported by noncontrolled and/or retrospective studies [48]. Immune depression with cyclophosphamide [49] and/or with anti-CD20 antibodies is a more recent approach presenting contrasting results [50]. Moreover, a combination of both plasmapheresis and the administration of anti-CD20 antibodies have been reported [51]. Future studies determining DNase activity during the therapeutic approaches are needed in order to verify a direct relationship between therapeutic efficacy and DNases inhibition.

## 8. Conclusions

Several studies on SLE and LN pathogenesis suggest that, in both conditions, the removal of NETs is hampered because of the functional defects of DNases. Genetic mutations affecting *DNASE1*, *DNASE2,* and *DNASE1IL3,* or the presence of DNases inhibitory agents (and/or DNases-directed autoantibodies) could explain DNases functional impairment. All of these studies highlight the relevance of NET DNA and NETosis, as a whole, as a central pathomechanism directly implicated in the onset and progression of SLE and LN. On the basis of the reviewed studies, it is tempting to hypothesize that the blockade or the selective depletion of anti-DNase autoantibodies could be a potential novel therapeutic approach to prevent or halt SLE and LN progression. More in general, strategies aimed at reducing NET formation might have a similar impact on the progression of SLE and LN. It is an approach that today can be envisioned thanks to the identification, using high-content screening technology [47], of clinical compounds able to prevent NET formation. Finally, recombinant DNases could also have a critical role to play in monogenic SLE.

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
