# Peer review of "Neutrophil Extracellular Traps-DNase Balance and Autoimmunity"

_cells, 2021, doi:10.3390/cells10102667_

Round 1
Reviewer 1 Report
This manuscript is a short review about impairment of digestion of DNA and SLE or LN.
I think this review is generally well written and appropriate for Special Issue "Neutrophil Extracellular Traps: From Host Defence to the Pathophysiology of Disease".
I would just propose to add one paragraph to discuss about treatment. Although they said in conclusion that “it is tempting to hypothesize that blockade or selective depletion of anti-DNase autoantibodies could be a potential novel therapeutic approach to prevent or halt SLE and LN progression”, how is its impact on vulnerability to infection. How is its possibility of the treatment for other autoimmune or inflammatory diseases? Balance of digestion and formation of DNase may be important.
Author Response
Rebuttal Letter
ID Manuscript cells-1381515
Title: Neutrophil Extracellular Traps-DNase balance and autoimmunity.
Reviewer 1
This manuscript is a short review about impairment of digestion of DNA and SLE or LN.
I think this review is generally well written and appropriate for Special Issue "Neutrophil Extracellular Traps: From Host Defence to the Pathophysiology of Disease".
[A] We may thank the Reviewer for the positive comment
I would just propose to add one paragraph to discuss about treatment. Although they said in conclusion that “it is tempting to hypothesize that blockade or selective depletion of anti-DNase autoantibodies could be a potential novel therapeutic approach to prevent or halt SLE and LN progression”, how is its impact on vulnerability to infection. How is its possibility of the treatment for other autoimmune or inflammatory diseases? Balance of digestion and formation of DNase may be important.
[A] We agree and we add the Paragraph 6 to face the treatment issue. We recently published an exhaustive Review, so we largely refer to the mentioned Review for the Pharmacological aspects (Front Med (Lausanne). 2021; 8: 614829.)
Reviewer 2 Report
This report is well described and interesting to the readers who concerns about SLE pathogenesis. I think howevere there are a few concerns.
- Line 161. After "dsDNA", isn't "antibodies" omitted?
- The authors should add a brief description of whether NETs formation is enhanced in SLE neutrophils, if possible. Because the authors mentioned the suppression of NETs formation as a therapeutic strategy in the conclusions, I think it would be better to mention a little about the NETs formation in SLE in the introduction or somewhere.
Author Response
Rebuttal Letter
ID Manuscript cells-1381515
Title: Neutrophil Extracellular Traps-DNase balance and autoimmunity.
Reviewer 2
This report is well described and interesting to the readers who concerns about SLE pathogenesis. I think howevere there are a few concerns.
Line 161. After "dsDNA", isn't "antibodies" omitted?
[A] Many thanks, we added it
The authors should add a brief description of whether NETs formation is enhanced in SLE neutrophils, if possible. Because the authors mentioned the suppression of NETs formation as a therapeutic strategy in the conclusions, I think it would be better to mention a little about the NETs formation in SLE in the introduction or somewhere.
[A] We agree. We mentioned it at the beginning of paragraph 2
Reviewer 3 Report
Overall this manuscript provides a useful although brief overview of the contributions of NETs to autoimmunity, particularly in SLE and lupus nephritis (LN). However, aspects of the writing require attention for it to reach publication quality – in some areas it appears to have been lifted from a PhD thesis with minimal alteration.
Corrections:
Lines 36-38: I am not sure that infections in patients with defective NADPH oxidase can be claimed as definitive evidence of a role for NETs in combatting infection. It is also likely to reflect a role for NADPH oxidase in the oxidative burst and microbial killing. This sentence should be tempered to reflect that.
Line 41 – remove ‘only’
Lines 43-45 – provide a reference with evidence that DNA and proteins together have hapten-like activity
Line 45 – This sentence needs attention for the quality of the English – e.g. ‘context’ not ‘contest’
Line 56 – It would be helpful to explain the significance of reduced C3/C4 levels in this context – what does this indicate?
Line 65 – ‘reputed’ not ‘deputed’
Line 66-73 – remove references to Chapters
Line 81 – DNA ‘as’ part of ‘a’ chromatin strand
Line 85/86 – it is unclear how anything stimulates the formation of an antibody against itself. This requires more detailed explanation – is it to do with breaking tolerance or hapten activity for example?
Line 85 – perhaps use another word for ‘plastic’ and ‘potential’ rather than ‘potentiality’
Line 97 – include ‘while’ in between ‘DNA,’ and ‘chromatin’
Line 109 – perhaps Inokuchi et al J Immunol 2020 204: 2088-97 is a better reference to use re DNAse1L3 production by myeloid cells than the review article cited here
Line 117 – did these patients have higher levels of circulating DNA? There seems to be a step missing between DNAse activity and antibody production
Line 119 – resembling ‘those in’ SLE patients
Line 163 – 20% of LN patients had ‘a 50% reduction in’ DNAse activity
Line 174 – ‘decreased’ instead of ‘decrease’
Author Response
Rebuttal Letter
ID Manuscript cells-1381515
Title: Neutrophil Extracellular Traps-DNase balance and autoimmunity.
Reviewer 3
Overall this manuscript provides a useful although brief overview of the contributions of NETs to autoimmunity, particularly in SLE and lupus nephritis (LN).
[A] We many thank the Reviewer for the positive comment
However, aspects of the writing require attention for it to reach publication quality – in some areas it appears to have been lifted from a PhD thesis with minimal alteration.
[A] We revised the entire Manuscript. We kindly ask the Reviewer to provide the Ref of the PhD thesis he mentioned, in case he’d continue to note the overlap.
No PhD students of our department had “NETs” as doctoral project, therefore we are really sorry but we don’t know how to further reduce the supposed overlap without knowing the source.
Our primary interest would be to recognise the work of a Doctoral Student mentioning her/him, limiting the overlap of course.
Corrections:
Lines 36-38: I am not sure that infections in patients with defective NADPH oxidase can be claimed as definitive evidence of a role for NETs in combatting infection. It is also likely to reflect a role for NADPH oxidase in the oxidative burst and microbial killing. This sentence should be tempered to reflect that.
[A] We tempered the sentence as well suggested
Line 41 – remove ‘only’
[A] We corrected
Lines 43-45 – provide a reference with evidence that DNA and proteins together have hapten-like activity
[A] We add a reference
Line 45 – This sentence needs attention for the quality of the English – e.g. ‘context’ not ‘contest’
[A] We corrected the typo
Line 56 – It would be helpful to explain the significance of reduced C3/C4 levels in this context – what does this indicate?
[A] We specified that C3/C4 represent common clinical marker of SLE activity
Line 65 – ‘reputed’ not ‘deputed’
[A] We corrected the typo
Line 66-73 – remove references to Chapters
[A] We removed them as suggested
Line 81 – DNA ‘as’ part of ‘a’ chromatin strand
[A] We corrected
Line 85/86 – it is unclear how anything stimulates the formation of an antibody against itself. This requires more detailed explanation – is it to do with breaking tolerance or hapten activity for example?
Line 85 – perhaps use another word for ‘plastic’ and ‘potential’ rather than ‘potentiality’
[A] We modified the sentence, trying to enforce the importance of nucleosome as antigen in SLE and LN. We strongly agree with Reviewer that the mechanisms leading to the development of autoantibodies represent, in general, the core problem of autoimmunity disease. However, we believe that discussion on that is out of the main topics of the present Mini-review.
Line 97 – include ‘while’ in between ‘DNA,’ and ‘chromatin’
[A] We added it
Line 109 – perhaps Inokuchi et al J Immunol 2020 204: 2088-97 is a better reference to use re DNAse1L3 production by myeloid cells than the review article cited here
[A] We added the suggested REF
Line 117 – did these patients have higher levels of circulating DNA? There seems to be a step missing between DNAse activity and antibody production
[A] We agree with the Reviewer on the importance of the data. We did not find it in the cited Manuscript
Line 119 – resembling ‘those in’ SLE patients
[A] We added it
Line 163 – 20% of LN patients had ‘a 50% reduction in’ DNAse activity
[A] We modified as suggested
Line 174 – ‘decreased’ instead of ‘decrease’
[A] We corrected the typo

Round 2
Reviewer 1 Report
I think the manuscript is revised appropriately.